# Multimodal Mobile Robotic Dataset for a Typical Mediterranean Greenhouse: The GREENBOT Dataset

**DOI:** 10.3390/s24061874

**Published:** 2024-03-14

**Authors:** Fernando Cañadas-Aránega, Jose Luis Blanco-Claraco, Jose Carlos Moreno, Francisco Rodriguez-Diaz

**Affiliations:** 1Department of Informatics, ceiA3, CIESOL, University of Almería, 04120 Almeria, Spain; fernando.ca@ual.es (F.C.-A.); jcmoreno@ual.es (J.C.M.);; 2Department of Engineering, ceiA3, CIESOL, University of Almería, 04120 Almeria, Spain

**Keywords:** dataset, LiDAR, stereo vision, SLAM, ROS

## Abstract

This paper presents an innovative dataset designed explicitly for challenging agricultural environments, such as greenhouses, where precise location is crucial, but GNNS accuracy may be compromised by construction elements and the crop. The dataset was collected using a mobile platform equipped with a set of sensors typically used in mobile robots as it was moved through all the corridors of a typical Mediterranean greenhouse featuring tomato crops. This dataset presents a unique opportunity for constructing detailed 3D models of plants in such indoor-like spaces, with potential applications such as robotized spraying. For the first time, to the authors’ knowledge, a dataset suitable to test simultaneous localization and mapping (SLAM) methods is presented in a greenhouse environment, which poses unique challenges. The suitability of the dataset for this purpose is assessed by presenting SLAM results with state-of-the-art algorithms. The dataset is available online.

## 1. Introduction

In recent years, the use of technology in agriculture has experienced significant growth, driven by improved productivity and resource optimization [1]. Technology is crucial for reducing the use of fertilizers and pesticides and for efficient water management [2]. Agricultural activities benefit considerably from technology, ranging from the application of micronutrients to the estimation of fruit quality and quantity, as well as the mechanical or electromagnetic removal of weeds [3]. Robots are presented as autonomous platforms that can carry out these tasks and can also be used to collect a large number of crop data continuously [4].

To carry out these tasks, it is essential to equip the robot with sensors to develop autonomous navigation algorithms in an agricultural environment. In the particular case of a greenhouse, localization requires excellent accuracy and odometry estimation, which is almost impossible to install in greenhouses [5]. This environment presents unique, challenging characteristics as repetitive patterns due to the lack of unique and consistent visual attributes of vegetation can be arduous, posing real distinctive challenges for algorithms. In addition, these natural environments present unique computer vision challenges due to subtle human-caused movements, extreme light variability due to the closed environment, seasonal crop changes (pruning, harvesting, etc.), and weather variability.

Large, open datasets are crucial for developing data-driven techniques and benchmarks in a new era of deep learning-based algorithms. In the literature, there are abundant datasets for urban environments (e.g., [6,7]), and to the authors’ knowledge, the most extensive datasets currently available for agricultural environments are the Rosario dataset [8], the Sugar Beets dataset [9], and the MAGRO dataset [10] for an apple orchard field. However, these datasets are focused on open-field agricultural environments where, on the one hand, localization and orientation may be less complex and, on the other hand, the permissible error is more significant than in closed environments, such as greenhouses, in particular, Mediterranean greenhouses. These greenhouses are located in warm or temperate regions with a plastic cover and without the usual use of heating systems. Usually characterized for being structured in irregular and narrow corridors (90–100 cm) with sandy soil (80% of farms), these types of greenhouses account for 92% of the total greenhouse area worldwide [11,12]. For this reason, they have been selected for this work.

This paper presents an innovative and dynamic framework for data collection and a dataset collected throughout the growing season in a typical Mediterranean greenhouse using a sensor-equipped forklift. The data collected include images from a stereo camera pointing forward, an IMU, and two 3D LIDARs. This dataset provides valuable information to advance the development of visual odometry. Notably, the data were acquired in a typical Mediterranean greenhouse with a tomato crop, which offers the opportunity to develop 3D models for the plants that can be especially useful, for example, for robotized spraying tasks, where the amount of plant protection product is applied while coordinating the relationship between the displacement speed of the robot and the spray speed [13,14]. It is important to remark that the tomato is one of the most important crops in southeastern Spain, occupying 17% of the greenhouse area in the province of Almería alone and accounting for 20% of its total crop production during the 2020/2021 season [15].

From a robotics perspective, recent advancements have introduced several robotic solutions for agriculture automation. Most of these developments are not technologically mature; however, some important companies in the agriculture machinery sector, such as John Deere, New Holland, and Case, and other emerging companies with applications in this sector, such as Naïo and DJI, have commercial products or one step away from commercialization. However, all of these developments have been conceived to work in the open field. For greenhouses, robot prototypes performing different tasks, such as weeding, harvesting, pruning, spraying, pest and disease monitoring, etc., have also been developed [16,17]. Since 1987, many researchers have worked on the development of robots for greenhouses, studying the different problems involved [18]. The AURORA project [19], for instance, introduced a resilient and cost-effective robot designed for greenhouse operations. This robot demonstrates the ability to navigate autonomously through diverse greenhouse environments. Another notable greenhouse automation initiative was described in [20], featuring a mobile robot equipped with a stereo vision system and a six-degree-of-freedom arm specifically designed for tomato cultivation in greenhouses. In addition, a service robot for sanitary control and localized dispensing of pesticides and fertilizers to plants in greenhouses using RGB-D cameras was presented in [21]. Other projects in Japan [22] have also received funding, contributing to the ongoing exploration and development of robotic solutions for greenhouse applications. It is important to notice that the authors of this paper have worked on three projects directly related to the use of mobile robots in greenhouses. In particular, ref. [23] presented the implementation and testing of navigation algorithms on the FITOROBOT mobile robot [13]. In the INVERSOS project [14], a multi-functional mobile robot was developed taking as a basis the experience provided by the FITOROBOT project, and in the AGRICOBIOT [24] project, another mobile robot for greenhouses was developed, but with the particular characteristic of working in a collaborative way close to humans.

The need for this dataset is demonstrated by the fact that there are a multitude of robots that can improve the navigation task. The sensor-equipped platform used in this article travels the same route in different conditions and on different days, capturing data with variations in lighting, crop condition, and weather. Along with the sensor data, details on the calibration of intrinsic and extrinsic sensor parameters and tools for data processing are provided. The data collected can be the subject of research on different topics, from optimizing the transport areas of materials inside the greenhouse in a more autonomous, fast, and efficient way to elaborating models on the behavior of climatic variables inside the greenhouse.

The primary achievements of this study can be outlined as follows: (i) the introduction of an innovative dataset focusing on an uncharted setting with complex and diverse characteristics (as far as the authors are aware, this is the initial investigation utilizing a dataset for the purpose of mapping and guiding mobile robots within greenhouses); (ii) the creation and release of an open-source process for collecting data, with plans to regularly enhance and expand the dataset; and (iii) the validation of the effectiveness of the dataset through the implementation of a cutting-edge SLAM algorithm. The dataset, along with its download instructions and the tools for data processing, can be accessed publicly in the dataset repository (University of Almería dataset “GREENBOT”: https://arm.ual.es/arm-group/dataset-greenhouse-2024/ accessed on 10 March 2024).

The rest of the paper is organized as follows. Section 2 presents a literature review, discussing the most relevant articles and focusing on works with application to agriculture. Section 3 presents the sensor-equipped platform and the calibration of the sensor. Section 4 presents how the data were acquired and the data structure. Section 5 evaluates the dataset with a novel algorithm to validate its quality, and finally, Section 6 ends with conclusions and future work.

## 2. Related Work

Datasets tailored to mobile robotics are important for tasks such as image-based localization, LIDAR odometry, and SLAM. Datasets can be classified according to whether they are synthetic (simulation of textures, objects, lighting, etc., from a virtual environment) or real (images or videos, point clouds, etc., from a real environment) or according to the type of scene. Table 1 summarizes the related datasets discussed in the rest of this section.

### 2.1. Synthetic Datasets

Starting from the works related to rural environments, Li et al. [25] propose a dataset based on a photorealistic environment, where they evaluated many SLAM techniques, even going into interiors with different RGB-D cameras. In the same line, Ros et al. [26] developed a dataset based on urban photos with stereo camera, building an urban 3D environment, and following the same technique, in [27], Giubilato et al. focused on mountainous environments. In the context of agricultural settings, in [28], Yang et al. introduced a composite dataset for assessing various SLAM methods across various plant varieties, camera angles, fruit markings, and more. Meanwhile, in [29], Sinha et al. constructed a simulated 3D vineyard environment to examine diverse mapping approaches. In [30], Lu and Young made a synthetic dataset for different agricultural areas divided into different plots of land for spraying or pruning of fruit trees, using a commercial camera.

### 2.2. Real Datasets

In these datasets, scenes can be categorized as indoor or outdoor, with outdoor scenes further classified based on whether they depict urban settings or other types of environments.

#### 2.2.1. Outdoors

In terms of environments, outdoor areas often present more difficulties due to changes in lighting conditions, weather, moving objects, etc. These can be further classified into urban or open-field environments, depending on the environment where the data were collected. The latter is particularly interesting for the present work because, although it is carried out in a greenhouse, which can be classified as indoor in agriculture, it is crucial to compare it with other open-field agriculture datasets, highlighting the non-existence of datasets for indoor agriculture environments, such as Mediterranean greenhouses.

**Table 1 sensors-24-01874-t001:** Summary of the surveyed datasets.

Dataset	Real/ Synthetic	Indoor/ Outdoor	Environment	Main Task	RGB	Depth	GPS	IMU	LiDAR	Ground Truth	Publicly Available
[25]	Synthetic	Indoor	Room	SLAM	Yes	Yes	No	No	No	No	No
[26]	Synthetic	Outdoor	Urban	Semantic segmentation for navigation	Yes	Yes	No	No	No	Pose	Yes
[27]	Synthetic	Outdoor	Mountain	SLAM	No	No	Yes	Yes	Yes	Pose	Yes
[28]	Real	Outdoor	Open field	SLAM	Yes	Yes	No	No	No	Semantic label	No
[30]	Synthetic	Outdoor	Different open field	Semantic segmentation for recognise	Yes	Yes	No	No	No	No	Yes
[7]	Real	Outdoor	Urban	SLAM	Yes	No	Yes	No	Yes	Pose	Yes
[6]	Real	Outdoor	Urban	SLAM	Yes	Yes	Yes	Yes	Yes	Pose	Yes
[31]	Real	Outdoor	Urban	SLAM	Yes	No	Yes	No	Yes	Pose	Yes
[32]	Real	Outdoor	Urban	SLAM	No	No	Yes	Yes	Yes	Pose	Yes
[33]	Real	Outdoor	Lake and rivers	Measuring	No	No	Yes	Yes	Yes	Pose	No
[34]	Real	Outdoor	Rivers	SLAM	Yes	No	Yes	Yes	No	No	Yes
[35]	Real	Outdoor	Various open field	Semantic segmentation for navigation	Yes	Yes	No	No	No	Pose	Yes
[36]	Real	Outdoor	Open field	Pest detection	Yes	Yes	No	No	No	No	No
[8]	Real	Outdoor	Soybean field	SLAM	Yes	Yes	Yes	Yes	No	Pose	Yes
[9]	Real	Outdoor	Sugar beet field	Weeding	Yes	Yes	Yes	Yes	Yes	Pose	Yes
[10]	Real	Outdoor	Apple field	SLAM	Yes	Yes	Yes	No	Yes	Pose	Yes
[37]	Real	Outdoor	Building	SLAM	No	No	Yes	Yes	Yes	Pose	Yes
[38]	Real	Indoor	House	SLAM	Yes	No	No	No	No	Pose	No
[39]	Real	Indoor	Laboratory	SLAM	Yes	Yes	No	No	No	Pose	Yes
[40]	Real	Outdoor	Laboratory scale greenhouse	Navigation techniques	Yes	No	No	No	No	No	No
[41]	Real	Indoor/ Outdoor	Mountain and building	SLAM	Yes	No	Yes	Yes	No	Pose	No
GREENBOT	Real	Indoor	Mediterranean greenhouse	SLAM	Yes	Yes	No	Yes	Yes	Pose	Yes

Urban environments

In [7], Maddern et al. presented a dataset for a university campus where data were collected using six RGB cameras, a LiDAR, and a GPS. In the study by Geiger et al. [6], data from an urban area were gathered using high-resolution color and greyscale stereo cameras, a Velodyne 3D laser scanner, and a high-precision GPS/IMU mounted on ground vehicles. In [31], Majdik et al. conducted a study where data were acquired by an aerial micro-vehicle flying at low altitudes over urban streets, equipped with high-resolution cameras, GPS, and IMU. In [32], Blanco-Claraco et al. introduced a dataset captured in Málaga using a Citroen C4 car equipped with a high-resolution stereo camera, recording data at a rate of 20 Hz over a distance of 36.8 km.

Open field environments

With the help of human-operated vehicles, in [33], Bandini et al. collected data of the river and lake vicinity over three years using a 2D LiDAR, RGB camera, GPS, and IMU. Similarly, Miller et al. [34] managed to gather data from the Sangamon River using a canoe equipped with an RGB camera, IMU, and GPS. Focusing on agriculture in open fields, in [35], de Silva et al. classified data obtained through an agricultural robot on a beet plantation for three months, acquiring data from an RGB-D camera, IMU, GPS, and LiDAR. Moreover, there are also articles addressing agricultural tasks unrelated to odometry, mapping, or navigation. In [36], Liu et al. presented a dataset for pest or harmful insect detection, and another study conducted fruit and plant detection with an RGB multi-spectral camera [42]. The related existing works with the most impact in agriculture focused on the dataset collected in Rosario [8] by Pire et al. and the Sugar Beets dataset [9] by Chebrolu et al., where, with the help of mobile robots, they could map using blending techniques based on georeferenced images and SLAM algorithms. These works focused on weed detection and mapping of different types of crops with an IMU, GPS, and LiDAR. Finally, in [10], Marzoa et al. presented a dataset using a robot with autonomous navigation, displaying images with RGB-D cameras and 3D LiDAR on a row of apple trees, closing the control loop through a SLAM algorithm.

#### 2.2.2. Indoors

Although most datasets were gathered outdoors, numerous studies on indoors datasets exist [43]. In this work, after conducting an exhaustive search, datasets are divided into buildings and greenhouses. However, as mentioned above, there is a lack of work analyzing datasets within greenhouses. In particular, to the authors’ knowledge, there is none for Mediterranean greenhouses.

Building environments

Depending on the scheme to be recorded, these datasets can adopt different types of complexity. Suppose the data are recorded in indoor environments, such as rooms. In this case, they are usually easier to recompile and process as they do not experience changes in lighting or significant differences related to seasonal changes. In addition, these environments are constructed of objects that the user can easily place. In this case, there are a multitude of datasets that recompile authentic interiors. For example, for a regular house, Kirsanov et al. in [38] presented a SLAM technique based on RGB cameras using different robotic platforms; in [37], Karam et al. proposed a drone equipped with a stereo camera and an IMU that acquires data from the interior of an industrial building. However, the variation of light, natural light, or the need to know the whole environment can lead to a disorientation of the robot, producing an erroneous map. In this case, Lee et al. in [39] contemplated the different casuistics that a service robot can experience when it navigates in interiors using stereo camera, 2D LiDAR, etc.

Agriculture environments

Localization by means of GNSS is fundamental for navigation with mobile robots [44], although weak signal reception inside greenhouses leads to many methods exploiting alternatives, such as rail-based locating [45] with a simple camera or beacons, such as QR codes by Feng et al. [20]. To date, most of the work on robots in greenhouses has been focused on (i) weed inspection and spraying and (ii) vegetable harvesting tasks. Regarding datasets, in [40], Martin et al. presented an unpublished proprietary small ROS dataset for data collection and mapping using different techniques with RGB cameras. Similarly, semantic mapping was performed in [46], where Matsuzaki et al. addressed contextual navigation in a laboratory scale greenhouse with agricultural robots with IMUs and different stereo cameras, using the CamVid dataset [41]. Other recent datasets have focused on detailed point clouds for phenotyping plants [47].

These works do not provide a detailed dataset that can be used to evaluate different SLAM techniques. The proposed work is entirely novel, as there is no dataset that employed two widely used LiDAR models and cameras navigating through different crop rows inside a greenhouse.

## 3. Platform Description

The platform used to record this dataset was designed at the University of Almería (with the support of the Regional Ministry of Economic Transformation, Industry, Knowledge and Universities and the European Regional Development Fund (FEDER) with the projects UAL2020-TEP-A1991 and PY2020_007-A1991) (Figure 1). It has a base size of 900 × 500 × 700 mm with a load capacity of 180 kg, equipped with four wheels and two handles for manual operation. The sensors are mounted on top of it, taking into account the specific constraints of each sensor:Bumblebee stereo camera BB2-08S2 (Santa Cruz, CA, USA) (https://www.flir.com/support/products/bumblebee2-firewire/ accessed on 10 March 2024): It has two lenses for capturing synchronized stereo images. It connected to the PC through a FireWire IEEE 1394 interface, providing a data transmission speed of 800 Mb/s. The stored data has a resolution of 1032 × 776 pixels. The horizontal field of view is 97°, and the vertical field of view is 66° within a range of 0.3 m to 20 m. Recording was performed at 10 Hz, with a maximum frame rate of 20 fps. This camera was placed on the front so that the images captured were ideal for obstacle identification for a robot. The collected images were stored in .bag format (ROS storages).Velodyne VLP16 LiDAR (San Jose, CA, USA) (https://velodynelidar.com/products/puck/ accessed on 10 March 2024): Velodyne’s LiDAR VLP16 has a maximum range of 100 m; a 360° horizontal and 30° vertical field of view was used. It was placed at the top of the runway to have the whole field of view available, except for the operator. Data were recorded with a frequency of 10 Hz.Ouster OS0 LiDAR (San Francisco, CA, USA) (https://ouster.com/products/hardware/os0-lidar-sensor accessed on 10 March 2024): Data were also collected from the Ouster model OS0. This is a 32-channel LiDAR with a 360° horizontal and 90° vertical field of view, ranging between 0.3 m and 50 m. The data rate was also left at the default value of 10 Hz, including the onboard 6-axis IMU. This sensor was placed in a structure 20 cm lower than the Velodyne, obstructing its view and, hence, only collecting data with an effective horizontal field of view of 275°. The point cloud output of both LiDARs were stored together with the .bag mentioned in the stereo camera, accessed with the different tools in ROS.

Because of the need for timestamp synchronization of all data from these sensors, it was necessary to install a mechanism that sets a fixed clock cycle. One of the most common techniques is to make a timestamp with the data collected by a GPS so that, always taking the same reference, synchronization is ensured [48]. For this dataset, the NovAtel GNSS, model IMU-IGM-A1 (https://novatel.com/support/span-gnss-inertial-navigation-systems/span-combined-systems/span-igm-a1 accessed on 10 March 2024) was installed, together with antenna ANTCOM 42G1215 (https://novatel.com/products/gps-gnss-antennas/compact-small-gnss-antennas/42g1215a-xt-1 accessed on 10 March 2024).

### 3.1. Computing Platform

Greenhouses often have difficult environmental conditions, with high temperatures and humidity [49]. Additionally, power for sensors and computers must be supplied from a battery that balances versatility and space. Considering this, the platform was equipped with a HISTTON computer featuring an Intel i7-8550U processor (Santa Clara, CA, USA) (4 GHz), an Intel UHD 620 graphics card with 24 CUDA cores, and 32 GB of DDR4 RAM. This equipment was selected for its operating temperature range of 0–70 °C and humidity range between 0 and 85%. It only consumes 15 W, resulting in minimal power consumption to consider for battery capacity. This computer was equipped with two disk partitions to use both Ubuntu 20.04 with ROS Noetic and Ubuntu 22.04 with ROS 2 Humble, providing the opportunity to work with both versions.

### 3.2. Calibration

#### 3.2.1. Intrinsic Parameters

The intrinsics of the stereo camera were determined with the kinect-stereo-calib application from the Mobile Robot Programming Toolkit (MRPT) [50], including the focal lengths (fx,fy), the optical centers (cx,xy), the pin-hole distortion parameters (k1,k2,p1,p2), and the left-to-right relative pose. Calibration was performed using a 7 × 10 classic checkerboard target pattern, capturing 20 pairs of images. The calibration file is available online. An example of the calibration tool, showing the result, is shown in Figure 2.

#### 3.2.2. Extrinsic Parameters

Extrinsic parameters define the relative poses of the different sensor frames of reference. The origin of coordinates is defined at the Velodyne, from which the other sensor poses are defined, as follows and shown in Figure 1:Stereo camera: *x* = right; *y* = down; *z* = forward.LIDARs: *x* = forward; *y* = left; *z* = up.

The Velodyne sensor is taken as a local frame of reference for the platform.

## 4. GREENBOT Dataset

This section describes the environment, data acquisition procedure, download method, and usage instructions.

### 4.1. Greenhouse and Crop

The experiments took place at the Agroconnect facilities (which received cofunding from the Ministry of Science, Innovation, and Universities in collaboration with the European Regional Development Fund (FEDER) under the grant program for acquiring cutting-edge scientific and technological equipment (2019)) in the Municipal District of La Cañada de San Urbano, Almería, located at 36°50′ N and 2°24′ W, with an elevation of 3 m above sea level and a slope in the terrain of 1% in the northern direction (see Figure 3).

The greenhouse is in one of the most common styles in the region (Almería’s “raspa y amagado”), expands over 1850 square meters, and features a sturdy steel frame and a polyethylene covering. The greenhouse is arranged in an east–west ridge configuration to benefit from the natural ventilation from those two predominant wind directions in the region. A 2 m wide central pathway in the greenhouse serves as the main road and leads to eleven aisles on each side. The aisles on the north side are 2 m wide and 12.5 m long, while the aisles on the south side are 2 m wide and 22.5 m long. Radiating from this central aisle are narrower secondary pathways, each only one meter wide, facilitating the smooth movement of mobile robotic units. The facility has two sections with various advanced systems tailored for precise crop management. These systems include natural ventilation from above (zenithal) and along the sides (lateral), an integrated air heating and cooling network, infrastructure for CO_2_ enrichment, cutting-edge equipment for humidification and dehumidification, an advanced irrigation and fertilization system, as well as energy-efficient LED artificial lighting.

The greenhouse crop is tomato (*Lycopersicon esculentum*), and the plants are grown in coir bags in rows oriented from north to south with a slope of 1%, as shown in Figure 4.

### 4.2. Data Acquisition

The trajectory that was followed through the crop is shown in Figure 5. In order to distinguish the longer corridors from the shorter ones, the greenhouse was segmented into two sections, dividing it by the central corridor.

The operator would drive the platform at an average velocity of 0.83 m/s, make one pass until the end of each aisle and another pass on the way back, and then switch to the next row. In the east and west corridors, no passes could be made because it was impossible to pass through the support pillars of the greenhouse. This would have resulted in a loss of data, which is not essential as the robot would not trace trajectories through these areas. It should be noted that the ninth row in section B did not have plants at the time of recording this dataset. In order to carry out this data collection, the point clouds provided by the Velodyne and the Ouster LIDARs, the Ouster IMU, and the stereo images from the Bumblebee camera were stored in ROS 1 bag format.

In terms of data structure, there are a total of nine sequences, recorded between 5 October 2022 and 1 December 2022. Weekly recordings were made, subjected to different environmental conditions, lighting, ground changes, etc. The first two data files contain information from section B only (refer to Figure 5, as section A was temporarily inoperative). The third sequence contains section A, separated from section B. Finally, the remaining datasets contain a single rosbag that stores the entire greenhouse. Table 2 shows detailed and summarized information about the stored data, providing information about the weather on the day of recording and the height of a pilot plant measured manually. The path length of each sequence was obtained from the SLAM-based reconstructions, as detailed later, using the evo_traj tool [51]. Finally, the climatological variables of air temperature, humidity, and irradiance inside the greenhouse during the navigation were extracted from a database of the University of Almería and also incorporated into the dataset.

### 4.3. Data Structure

As already mentioned, the data were stored in rosbag format as it allowed storing the desired topics in the original format. This rosbag contains the ranging data from both LIDARs, the stereo images, and the IMU from the Ouster LIDAR.

Alternatively, LIDAR data were also provided in rawlog format (description of rawlog format https://docs.mrpt.org/reference/latest/robotics_file_formats.html accessed on 10 March 2024). This format can be easily parsed and published by ROS 2 Humble packages, such as mrpt_rawlog or the SLAM framework described later.

## 5. SLAM Suitability Assesment

This section assesses the applicability of the dataset to build maps using a LIDAR SLAM framework. In particular, we used mola_lidar_odometry from the MOLA framework [52]. The approach belongs to the family of LIDAR odometry methods (i.e., without loop closure) with voxel-based raw point cloud representation. Experimental results from both LiDARs are presented separately in the following subsections.

### 5.1. Mapping Velodyne VLP16

Figure 6 shows the result of the greenhouse mapped with the Velodyne VLP16 LiDAR for four different days. To aid visualization, false color has been used to reflect point height.

A significant visual difference in the state of the tomato plants can be observed in Figure 6a–d, guaranteeing the systematic mapping of the greenhouse. False LiDAR points are seen at the top of each image above the actual top surface of the greenhouse. These points are standard for VLP-16 LiDAR, which detects false points beyond the plastic surfaces of the greenhouse. However, these data are not qualified as design parameters, as they come from the reflection of the greenhouse plastic itself. Greenhouse soil is often uneven, with slopes varying between 1 and 2%, depending on the area. In this type of mapping, being able to obtain a terrain variation model is of vital importance for robotics [53]. As the output of the SLAM module, both an estimated 3D map of the environment and the estimated platform trajectory were obtained and then analyzed and plotted using evo_traj as shown in Figure 7.

It can be seen how the pattern of movement is repeated every day. It is essential to mention that the trials started with the same orientation, as can be seen in Figure 7a–d toward +x and +y. Each axis exhibits consistent behavior regardless of the day, although interesting patterns can be noticed along each path. To analyze this, the path followed by each axis on 30 November 2022 is shown in Figure 8.

The sub-graphs corresponding to the displacement in *x* and *y* correspond to the displacement in the 2D plane, but the graph in *z* shows data relevant to the investigation. Noise that comes from the deformed soil in the greenhouse is observed, as well as a slope corresponding to the slope of the terrain itself, validated by the algorithm itself. Finally, Figure 9 and Figure 10 show final orthogonal and plan views, respectively, of the result of the mapped 3D model, indicating the dataset’s quality.

### 5.2. Mapping-Ouster OS0

The same SLAM algorithm was applied to the Ouster LiDAR data, obtaining the results shown in Figure 11. In this case, the mapping is shown for the same days analyzed in the previous section. Similarly, Figure 12 shows the result of the trajectories recorded by the Ouster OS0.

A higher point cloud density can be observed, as this LiDAR has twice as many point cloud rings as the VLP-16. In the same way, each of the segmented axes is projected, exhibiting the same oscillatory behavior due to the uneven terrain.

The original and plan views are presented in Figure 13 and Figure 14, obtaining an excellent result and exceeding the dataset expectations.

## 6. Conclusions and Future Work

A novel dataset tailored for demanding natural agricultural settings is introduced in this paper to advance the progress of autonomous robots that can enhance agricultural operations. The paper elaborates on the data collection methodology and outlines the utilized approach. Furthermore, the dataset is verified using an innovative SLAM technique to ensure its functionality. The findings suggest that the data quality is adequate for integration into specific algorithms. The provision of open datasets, such as the one presented here, is crucial in promoting further research and deployment of autonomous robots in agricultural settings within greenhouses, given the distinctive features of these environments, such as significant variations in lighting, unpredictable weather conditions, and seasonal fluctuations.

This dataset provides an opportunity for the community to create and evaluate one’s own SLAM algorithms in a unique agricultural setting, focusing on intensive greenhouse farming. Accurately mapping, orienting, and navigating are crucial for various robot activities within a greenhouse, such as transporting produce crates, harvesting, spraying, and collecting data.

The proposed framework is intended to be used as a database, which is unique for now. Data will continue to be collected from the greenhouse as all these sensors will be implemented in two robots of the University of Almería’s own. This implementation aims to carry out numerous validations of the usefulness of the data obtained, in particular, a systematic comparison of the autonomous navigation of an Ackermann robot and a differential robot, research into cooperative work between robots in greenhouses, and research into collaborative robot-farmer work. It is important to note that it is intended to collect data more frequently by implementing robots, seeking a more extensive dataset.

Apart from installing these sensors in real robots, the possibility of installing RGB-D cameras on both sides of the robots will be explored to obtain detailed information about the crop. The implemented cameras will also be improved to make them more robust to sudden changes in lighting, crop growth, etc. All this will be implemented in ROS 2 Humble, providing the robot with the latest technology in mobile robotics. Finally, these models can be used for other tasks, so this exciting contribution can be used to obtain the semantic information of the crop type, determining a model of plant growth in height and width.

This paper aims to present and share a unique dataset with the potential to assist researchers in developing new SLAM-based techniques and in evaluating a totally novel environment, such as a greenhouse. It is outside the scope of this work to compare and analyze the performance of existing methods. The results described in Section 5 are provided as a reference that researchers can use as a starting point for future comparative evaluations.

## Figures and Tables

**Figure 1 sensors-24-01874-f001:**
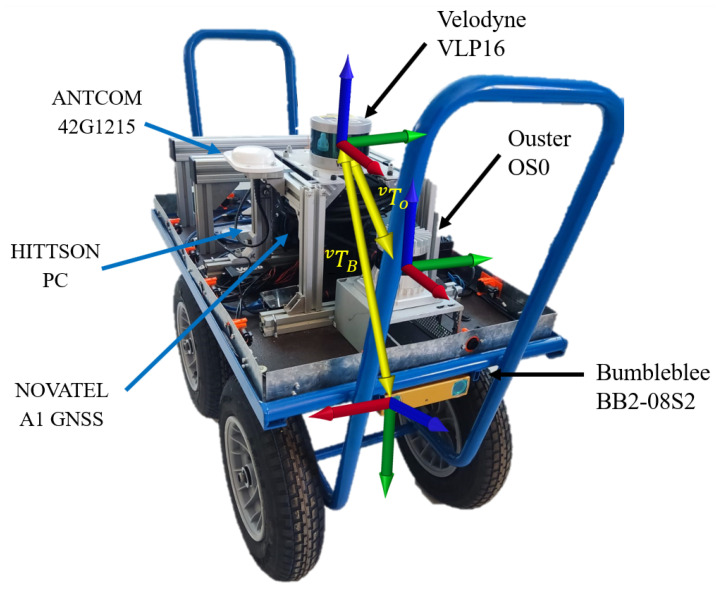
Sensor-equipped platform used to acquire data, equipped with a Bumblebee stereo camera, Novatel GNSS (Calgary, AB, Canada), ANTCOM 45G1215 (Torrance, CA, USA), a Velodyne VLP16 3D LiDAR, and an Ouster OS0 LiDAR with an IMU. Also, a map of reference of local systems for platform sensors is shown. The x-axis is represented in red, the y-axis in green, and the z-axis in blue.

**Figure 2 sensors-24-01874-f002:**
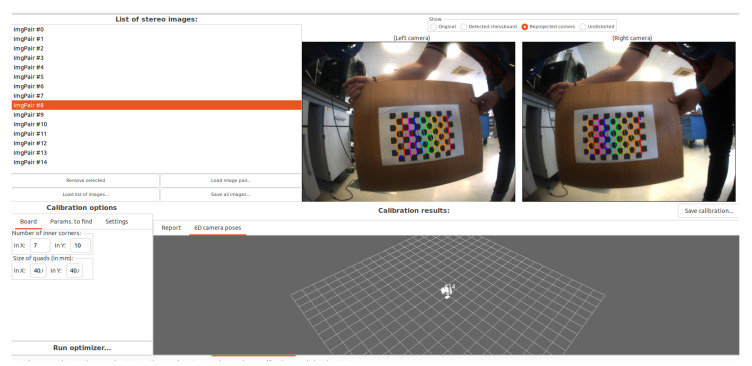
Screenshot of the intrinsic calibration application used for the stereo camera [50].

**Figure 3 sensors-24-01874-f003:**
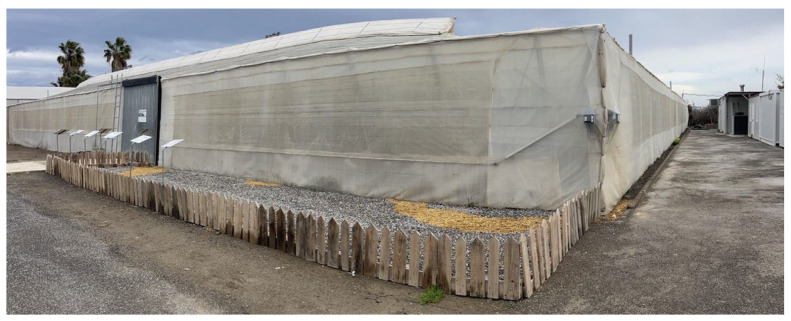
IFAPA experimental greenhouse.

**Figure 4 sensors-24-01874-f004:**
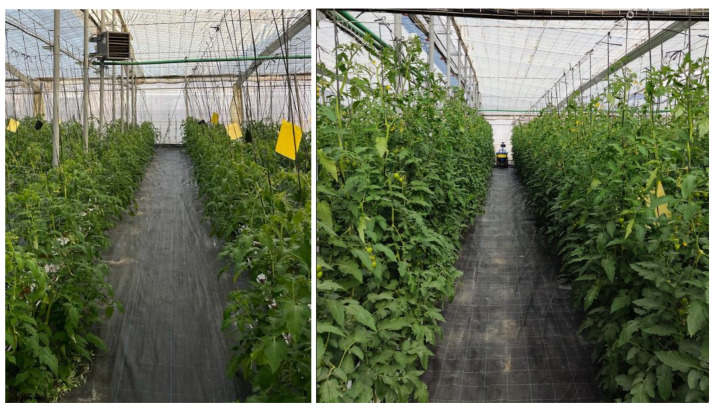
Example of two images acquired at the same position with different crop stages: the first one (**left**) after one month of plantation and the second one (**right**) after three months.

**Figure 5 sensors-24-01874-f005:**
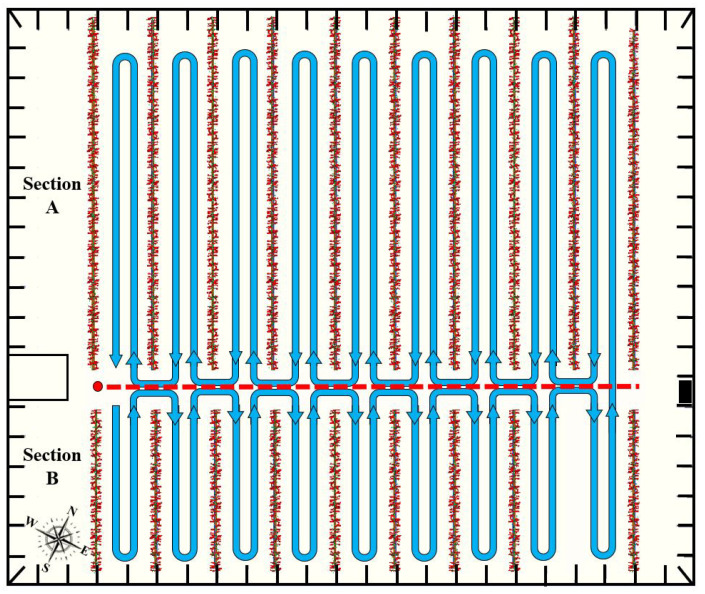
Trajectory taken during the data collection campaign in the greenhouse of the IFAPA center in Almería. The red dot indicates the start and end of the route.

**Figure 6 sensors-24-01874-f006:**
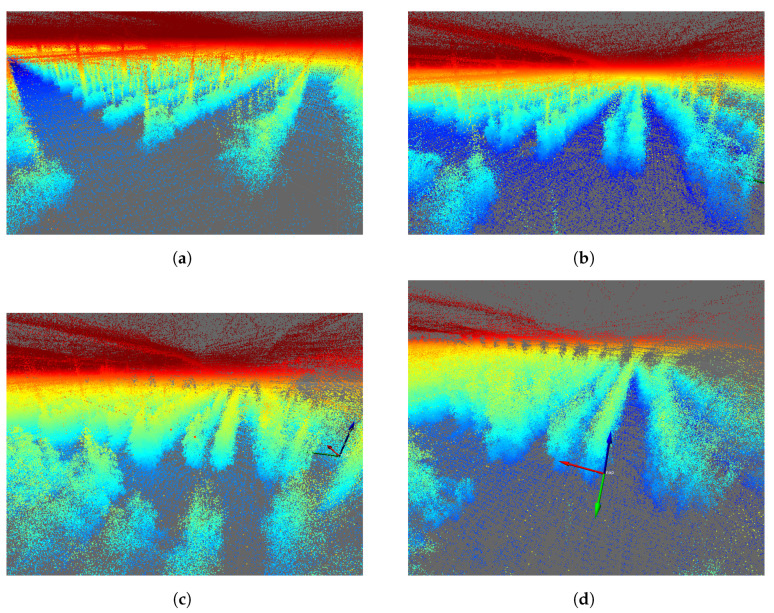
The 3D mapping of the IFAPA greenhouse with Velodyne VLP16 with a jet-type color intensity. (**a**) Velodyne MOLA Lidar odometry 2022-10-14. (**b**) Velodyne MOLA Lidar odometry 2022-10-26. (**c**) Velodyne MOLA Lidar odometry 2022-11-09. (**d**) Velodyne MOLA Lidar odometry 2022-11-30.

**Figure 7 sensors-24-01874-f007:**
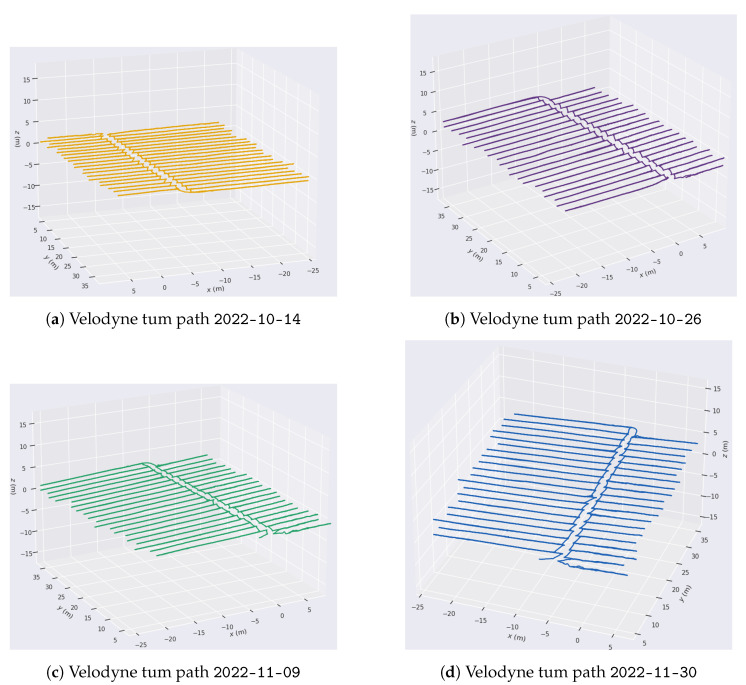
Path followed on different days with the Velodyne VLP16.

**Figure 8 sensors-24-01874-f008:**
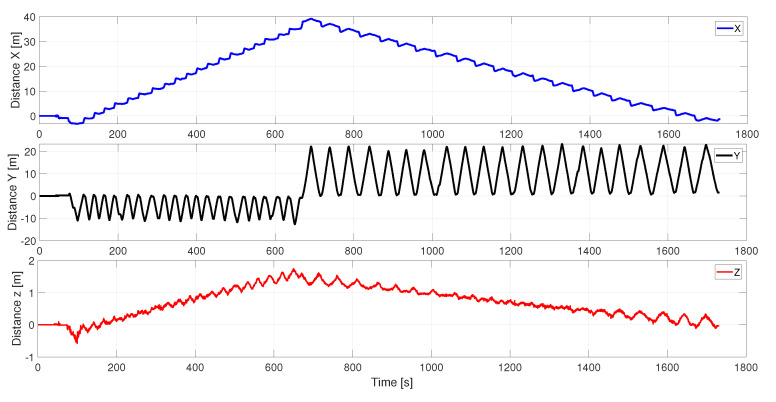
Trajectory estimated from the LiDAR data for sequence 2022-11-30.

**Figure 9 sensors-24-01874-f009:**
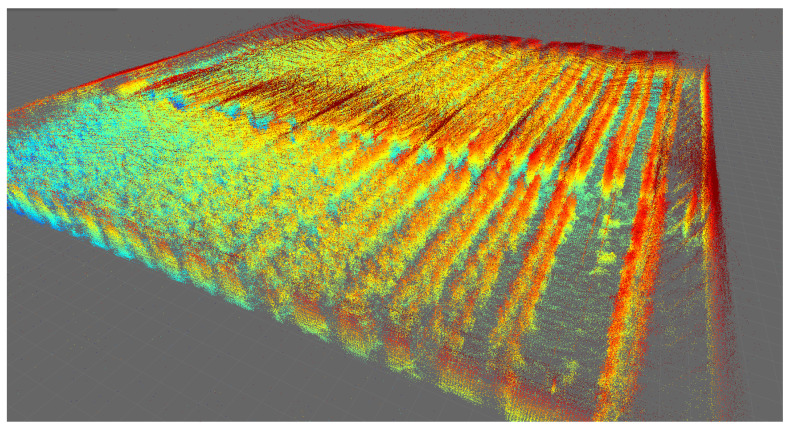
Mapping of the complete greenhouse, orthogonal view with Velodyne VLP16 color indicates point height above ground using the jet colormap).

**Figure 10 sensors-24-01874-f010:**
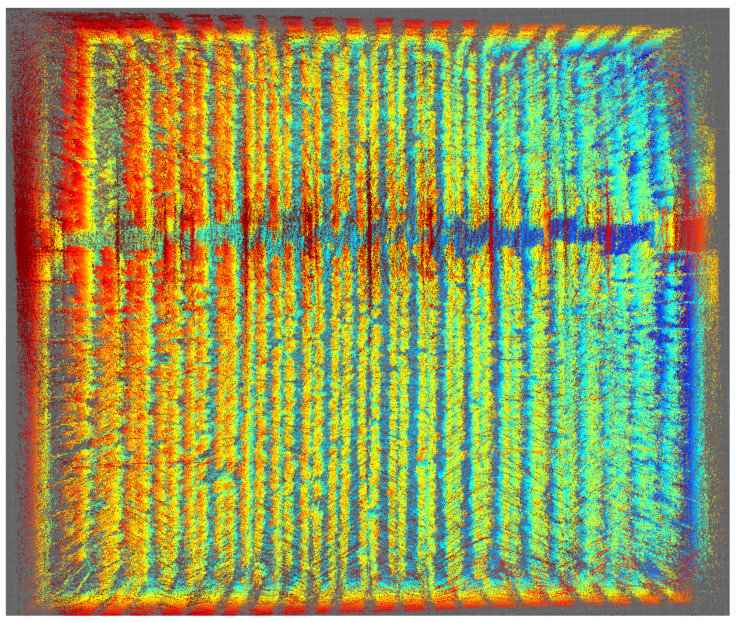
Mapping of the complete greenhouse, plan view with Velodyne VLP16 (color indicates point height above ground using the jet colormap).

**Figure 11 sensors-24-01874-f011:**
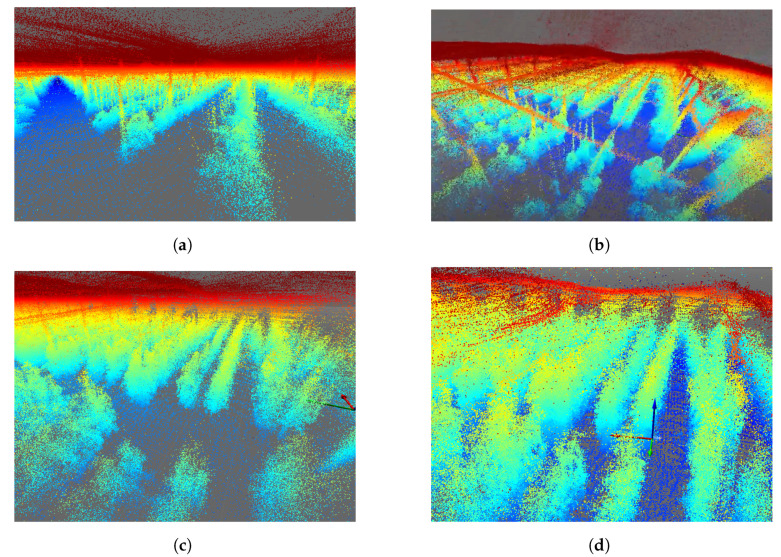
The 3D mapping of the IFAPA greenhouse with Ouster OS0. (**a**) Ouster MOLA Lidar odometry 2022-10-14. (**b**) Ouster MOLA Lidar odometry 2022-10-26. (**c**) Ouster MOLA Lidar odometry 2022-11-09. (**d**) Ouster MOLA Lidar odometry 2022-11-30.

**Figure 12 sensors-24-01874-f012:**
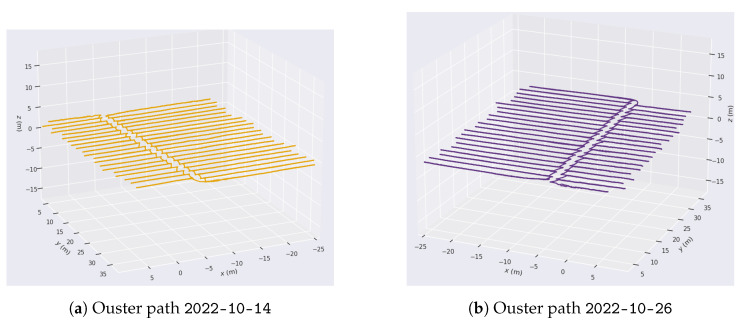
Path followed on different days with the Ouster OS0 with a jet-type color intensity.

**Figure 13 sensors-24-01874-f013:**
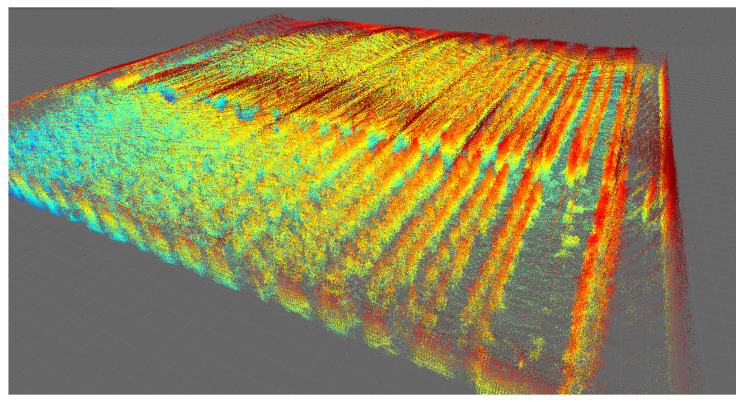
Mapping of the complete greenhouse, orthogonal view with Ouster OS0.

**Figure 14 sensors-24-01874-f014:**
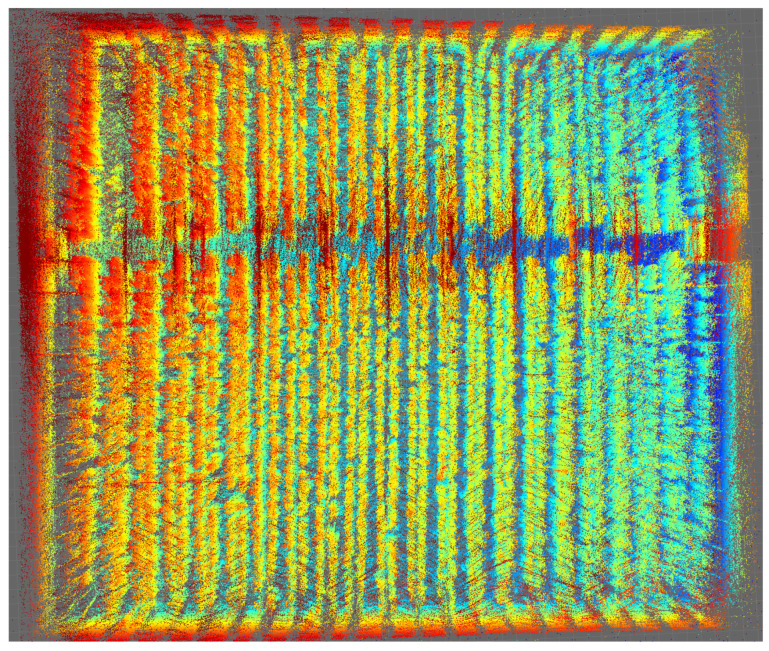
Mapping of the complete greenhouse, plan view with Ouster OS0.

**Table 2 sensors-24-01874-t002:** Description of each recorded segment.

Sequence	Length [m]	Duration [s]	Section	Greenhouse Condition	Description
2022_10_05	459.25	696	B	Temp: 27.01 °C Hum: 62.72% Ir: 106.7 W/m^2^	Sunny, morning plant height 0.96 m
2022_10_14	457.36	701	B	Temp: 23.78 °C Hum: 63.12% Ir: 79.0 W/m^2^	Cloudy, morning plant height 1.05 m
2022_10_19	1321.21	1432	A and B	Temp: 25.21 °C Hum: 75.83% Ir: 87.9 W/m^2^	Sunny, morning plant height 1.12 m
2022_10_26	1432.08	1463	A and B	Temp: 23.22 °C Hum: 60.09% Ir: 66.4 W/m^2^	Cloudy, morning plant height 1.35 m
2022_11_02	1233.87	1486	A and B	Temp: 15.84 °C Hum: 72.35% Ir: 70.7 W/m^2^	Cloudy, morning plant height 1.41 m
2022_11_09	1293.29	1532	A and B	Temp: 16.23 °C Hum: 62.7% Ir: 82.7 W/m^2^	Cloudy, morning plant height 1.53 m
2022_11_19	1332.58	1752	A and B	Temp: 17.45 °C Hum: 62.7% Ir: 64.36 W/m^2^	Sunny, morning plant height 1.60 m
2022_11_23	1428.45	1692	A and B	Temp: 15.28 °C Hum: 62.7% Ir: 69.4 W/m^2^	Cloudy, morning plant height 1.73 m
2022_11_30	1440.32	1730	A and B	Temp: 16.05 °C Hum: 62.7% Ir: 74.4 W/m^2^	Cloudy, morning plant height 1.85 m

## Data Availability

The dataset presented in this work is available online at https://arm.ual.es/arm-group/dataset-greenhouse-2024/ (accessed on 10 March 2024).

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
