# Peer review of "Multimodal Mobile Robotic Dataset for a Typical Mediterranean Greenhouse: The GREENBOT Dataset"

_sensors, 2024, doi:10.3390/s24061874_

Round 1
Reviewer 1 Report
Comments and Suggestions for Authors
This paper presents an innovative dataset tailored for challenging agricultural settings, where achieving precise localization is paramount. Collected using a mobile platform equipped with sensors commonly found in mobile robots, the dataset covers all corridors of a typical Mediterranean greenhouse with tomato crops. This dataset offers a unique opportunity to create detailed 3D models of plants in indoor-like spaces, with potential applications like robotized spraying. To the authors' knowledge, this marks the first presentation of a dataset suitable for testing SLAM methods in a greenhouse environment, posing unique challenges. The dataset's suitability for this purpose is assessed by showcasing SLAM results using state-of-the-art algorithms. In summary, the paper could be enhanced in the following areas:
1. The abstract contains several spelling errors, such as "challengues," requiring correction.
2. The abstract should explicitly state the research motivation, including comparisons with similar datasets.
3. The Abbreviations section before the references needs revision and should be presented in a table format.
4. The referencing style in Section 2 requires clarification, avoiding the format of citing literature + verb and emphasizing author information.
5. Section 2's structure needs further refinement, integrating key terms like Dataset, LiDAR, Stereo vision, SLAM, ROS, and emphasizing the research motivation. Additionally, the conclusion's future work should provide more detailed insights.
Comments on the Quality of English Language
Moderate editing of English language required.
Author Response
Responses to Reviewer 1
We appreciate your thorough review and valuable feedback. Below are our detailed responses to each of your comments, which have undoubtedly contributed to enhancing the paper’s quality.
-
The abstract contains several spelling errors, such as, challengues, requiring correction.
Thank you very much for your comment. The abstract has been revised and these errors have been corrected.
-
The abstract should explicitly state the research motivation, including comparisons with similar datasets.
Thanks for your suggestion. The abstract has been updated to clarify that one of the main motivations for this research is localization and/or SLAM in these particular agricultural environments, which mix together outdoor and indoor features.
Regarding the comparison with other similar datasets in the abstract, we explicitly say:
For the first time, to the best knowledge of authors, a dataset suitable to test Simultaneous Localization and Mapping (SLAM) methods is presented in a greenhouse environment, which pose unique challenges.
A detailed comparison with other datasets cannot fit in the abstract and is hence delayed until section 2.
-
The Abbreviations section before the references needs revision and should be presented in a table format.
Thank you very much for your comment. The abbreviations have been revised and summarized in the new Table 3.
-
The referencing style in Section 2 requires clarification, avoiding the format of citing literature + verb and emphasizing author information.
Thank you for your comment. This recommendation has been observed in the revised text.
-
Section 2’s structure needs further refinement, integrating key terms like Dataset, LiDAR, Stereo vision, SLAM, ROS, and emphasizing the research motivation. Additionally, the conclusion’s future work should provide more detailed insights.
During the revision, we performed several changes in this section to further clarify the research motivation of both, related works, and ours. Please, refer to colored text in the revised manuscript for changes.
As for the comment related to the conclusion, the article sets out various future works, although more details have been provided in the work that will be implemented in the short term. The paragraph introduced in the new version of the paper is attached below:
This implementation aims to carry out numerous validations of the usefulness of the data obtained, in particular a systematic comparison of the autonomous navigation of an Ackermann robot and a differential robot, research into cooperative work between robots in greenhouses and research into collaborative robot-farmer work.
Reviewer 2 Report
Comments and Suggestions for Authors
In this study, the authors collected a dataset for challenging agricultural conditions in greenhouses. The collected data contain varying modalities such as RGB-D images, IMU measurement and point clouds. The sensors are mounted on a mobile platform which can be used later for autonomous navigation research. The dataset has characteristics which are not present in other existing datasets, thereby it will serve as a challenging benchmark for different research tasks related to AI for agriculture. The manuscript is well-written, and technical details are clearly presented. One suggestion is the authors may need to describe more about the image and IMU data other than the Lidar point clouds. The authors should also discuss which research tasks the collected dataset can be used for.
Author Response
Responses to Reviewer 2
In this study, the authors collected a dataset for challenging agricultural conditions in greenhouses. The collected data contain varying modalities such as RGB-D images, IMU measurement and point clouds. The sensors are mounted on a mobile platform which can be used later for autonomous navigation research. The dataset has characteristics which are not present in other existing datasets, thereby it will serve as a challenging benchmark for different research tasks related to AI for agriculture. The manuscript is well-written, and technical details are clearly presented.
Thank you for the detailed evaluation and insightful suggestions. We have carefully considered each comment and integrated relevant responses to greatly improve the quality of the paper.
-
One suggestion is the authors may need to describe more about the image and IMU data other than the Lidar point clouds.
Thank you for these important comments.
It is true that despite we claim the dataset “multimodality” due to using different sensors, the experimental validation (SLAM) was solely carried out with LiDAR. This is mainly because we experimentally found that due to the way pointclouds are distributed in these scenarios it was actually quite challenging to get a LiDAR SLAM framework to work cleanly on the dataset and produce a decent map. We achieved it with the MOLA SLAM framework, but many open-sourced LiDAR SLAM frameworks failed, effectively demonstrating how challenging the dataset is. However, we decided not to put the focus on any particular SLAM method or any SLAM comparison in the paper, and keep it on the dataset itself.
On the images, running vision-based SLAM would take us much longer than the time provided for this review.
We believe that the basic properties of both, images and IMU readings, are already described well in section 3. By the way, the reduced IMU data rate makes their readings not suitable for visual-inertial SLAM, hence it is not even mentioned in the text.
Regarding the synchronization of all these sensors, to ensure that all elements have the same timestamp it is necessary to synchronize them with a single clock. In our case, satellite synchronization has been chosen, so the satellite timestamp is obtained and marked as the clock timestamp. This paragraph (with a new reference) has been added on lines 224-227.
Due to the need for timestamp synchronization of all data from these sensors, it is necessary to install a mechanism that sets a fixed clock cycle. One of the most common techniques is to make a timestamp with the data collected by a GPS so that, always taking the same reference, synchronization is ensured [49]. For this dataset, the…
[49] Mañas-Álvarez, F.J.; Guinaldo, M.; Dormido, R.; Dormido, S. Robotic Park. Multi-Agent Platform for Teaching Control and 510 Robotics. IEEE Access 2023.
-
The authors should also discuss which research tasks the collected dataset can be used for.
Thank you for your comment. The Conclusions and future work Section has been rewritten in order to clarify this issue, from two points of view. On the one hand, detailing the agricultural tasks, susceptible robotic-based automation in greenhouses, for which there is high research interest (lines 369-373):
Accurately mapping, orienting, and navigating are crucial for various robot activities within a greenhouse, such as transporting produce crates, harvesting, spraying, and collecting data.
and on the other hand, research tasks in which authors are currently involved (lines 376-380):
This implementation aims to carry out numerous validations of the usefulness of the data obtained, in particular a systematic comparison of the autonomous navigation of an Ackermann robot and a differential robot, research into cooperative work between robots in greenhouses and research into collaborative robot-farmer work.
Reviewer 3 Report
Comments and Suggestions for Authors
- In the review, it is advisable to list or systematize the existing types of greenhouses in the Mediterranean by quantitative indicator, for example area
- Line 159 - it is advisable to provide references to literature
- Line 273 -274 how greenhouse design will affect data acquisition. You need to modify the greenhouse to suit your data acquisition technology?
- Figure 6 It is advisable to indicate in the figure the numerical period of the color intensity scale
- Line 310.This parameter may be requirements for greenhouse design?
- Does the work plan for robotization and creation of digital parameters include data on the early infection of crops with pathogens?
There are already operating greenhouses and greenhouses that are at the design stage. The existing greenhouses are unified.To leverage your digital data acquisition technology, I would recommend considering three types of robotic process automation:
1. For existing greenhouses, taking into account their classification and breakdown into parameters: aisle length area, crop height, etc.
2. Write down the initial technical data for greenhouse developers (greenhouse design), taking into account the robotization features of this crop
3. Register the initial technical data for modernizing the design of existing types of greenhouses
Author Response
Responses to Reviewer 3
We appreciate your thorough review and valuable feedback. Below are our detailed responses to each comment, which have undoubtedly contributed to enhancing the paper’s quality.
-
In the review, it is advisable to list or systematize the existing types of greenhouses in the Mediterranean by quantitative indicator, for example area.
Thank you for your suggestion. The introduction has been amended as follows:
These greenhouses are those located in warm or temperate regions with a plastic cover and without the usual use of heating Systems. Usually characterized for being structured in irregular and narrow corridors (90-100 cm) with sandy soil (80% of farms). These kinds of greenhouses accounts for 92% of the total greenhouse area world-wide [11,12]. For the latter reason, they have been selected for this paper.
-
Line 1–9 - it is advisable to provide references to literature.
Thank you for your comment. The quoted paragraph was not correctly worded and has therefore been reworded to make it clearer (9-10).
-
Line 273 -274 how greenhouse design will affect data acquisition. You need to modify the greenhouse to suit your data acquisition technology?
Thank you for this comment. This part of the greenhouse is not passable, so data from it must not be acquired. A clarification has been included in the text:
In the east and west corridors, no passes could be made due to the impossibility of passing through the support pillars of the greenhouse. This will result in a loss of data, which is not essential as the robot will not trace trajectories through these areas.
-
Figure 6 It is advisable to indicate in the figure the numerical period of the color intensity scale
Thank you for this comment. A sentence indicating the color intensity range has been added to the figures’ caption.
-
Line 310. This parameter may be requirements for greenhouse design?
Thank you very much for your comment. In this case the slope of the soil is between 1 and 2%, which is the norm for this type of greenhouse. When constructing these greenhouses, slopes of no more than 2% are sought.
-
Does the work plan for robotization and creation of digital parameters include data on the early infection of crops with pathogens?
Thank you for your interesting suggestion. This dataset can be used to design or test navigation techniques for example to go through the whole greenhouse or part of it in spraying tasks, but it has not been considered to include pest & disease data for this task as it is not the aim of the paper. Still, thanks for the comment because it opens up new lines of related work.
There are already operating greenhouses and greenhouses that are at the design stage. The existing greenhouses are unified. To leverage your digital data acquisition technology, I would recommend considering three types of robotic process automation:
1. For existing greenhouses, taking into account their classification and breakdown into parameters: aisle length area, crop height, etc.
2. Write down the initial technical data for greenhouse developers (greenhouse design), taking into account the robotization features of this crop.
3. Register the initial technical data for modernizing the design of existing types of greenhouses.
Thank you very much for your suggestions. We will take them in consideration in future works. For the manuscript being, we kept the focus as much as possible on the dataset itself which is the claimed contribution for the community.
Reviewer 4 Report
Comments and Suggestions for Authors
This paper presents the scenario and the process of data acquisition to generate several datasets in a greenhouse environment using a robotic platform.
The paper shows really interesting results. I would also recommend the authors to:
- Include the legend to interpret Figures 6, 9, 10, 11 correctly.
- Provide more details on how Figure 8 and Figure 13 are estimated.
With these changes and the corrections of english mistakes I would consider this work perfectly acceptable.
Comments on the Quality of English Language
Although good explanations are given, there are several English issues that must be arranged. I would recommend the authors to reword some sentences in:
- Section 1, line 22.
- Repetition of ‘unique’ in Section 1, line 27,28.
- Section 2.1, lines 112-115.
- The word classifiable in Section 2.2. line 120.
- Section 5.1, line 313.
- Section 5.1, line 315-316.
Author Response
Responses to Reviewer 4
This paper presents the scenario and the process of data acquisition to generate several datasets in a greenhouse environment using a robotic platform. The paper shows really interesting results.
Thank you for the detailed evaluation and insightful suggestions. We have carefully considered each comment and integrated relevant responses to greatly improve the quality of the paper.
1. Include the legend to interpret Figures 6, 9, 10, 11 correctly.
Thank you for this comment. A sentence indicating the color intensity range used has been added to the text.
2. Provide more details on how Figure 8 and Figure 13 are estimated.
Those figures are the X, Y, and Z coordinates of the vehicle estimated trajectory (an output of the SLAM module), plotted via the evo python3 package (the evo_traj command, in particular). The text cites the evo package it now explains better these facts.